# InV3st: Open-Vocabulary Object Detection and Instance Segmentation in 3D Crime Scenes

Michael Greza[*1], Florian Eichinger[1], Yi Wang[1], and Benjamin Busam[1]

[1]Technical University of Munich, Photogrammetry & Remote Sensing
{michael.greza, florian.eichinger, yi.wang, b.busam}@tum.de

## Abstract

Enabling an open-world vocabulary object detection and segmentation in 3D scenes is a current challenge in 3D computer vision. One of the application fields that can profit from this is crime scene investigation where digital twins of felonies are increasingly common. These scenes often-times encompass a very large amount of arbitrary objects. We propose a vision language model based processing pipeline that creates a latent-space representation of the full contents of a Gaussian splat scene through DINOv3/SigLIP2 feature extraction that can be queried with open-world vocabulary to find objects. To ease computational cost, the system operates in 2D image space and then segments found objects of arbitrary size within the 3D scene. Our pipeline is designed to work on cluttered, large scenes with many details. Results and their evaluation will be presented at Northern Lights Deep Learning 2026.

## 1 Introduction

Object detection in cluttered 3D scenes is a challenging task e.g. due to partial occlusions and, depending on scene size and number of objects, computational complexity. When applied to crime scene investigation, these issues can increase heavily. Crime scenes are often-times chaotic, messy and stretch over several rooms or even streets, especially in cases of felonies. Police units in the last decade began to conserve the state of crime scenes before they are cleaned to be able to reenter them in virtual reality. The goal is, inter alia, to have an as-realistic-as-possible digital twin for further scene inspection or sanity checks on testimonies.

An automated detection of objects within a digital twin of a crime scene can ease the work of detectives. Further, it would enable the creation of an inventory of the crime scene to match with other cases and find repeating patterns in present objects. Discrete-class-based object detection is not sufficient for this, as the type of object of importance from case to case can be manifold and not foreseeable. We propose an open-vocabulary deep-learning-based pipeline for the generation of latent space represen-

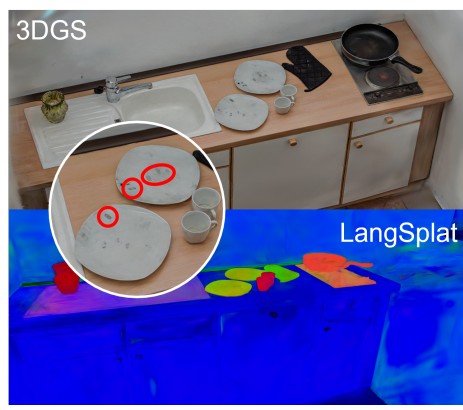

**Figure 1.** Examplary

tations of 3D scenes that is semantically searchable through a vision language model (VLM).

## 2 Related Research

Crime scene investigation within virtual reality is increasingly utilized by federal police units. As one early example, the Bavarian state police is researching and working on meshed 3D twins from imagery and LiDAR scans since 2012, now transitioning towards Gaussian Splats.

In recent years an increasing number of methods are proposed in the field of open-vocabulary 3D scene understanding, embedding 2D vision-language features into 3D representations. LangSplat [1] lifts multi-level CLIP [2] features extracted from SAM [3] segmentation-masks into 3D through feature compression, establishing a 3D Gaussian language field for open-vocabulary querying. However, its aggregation tends to blur semantics across large scenes and small or occluded objects, posing challenges in digital crime scene analysis, displayed in Figure 1.

Dr.Splat [4] and Occam's LGS [5] improve runtime efficiency by directly injecting language embeddings into individual Gaussians, enhancing per-point semantic consistency but lacking occlusion handling - especially posing a problem in unstructured and chaotic environments as crime scenes. VALA [6] adresses these issues by introducing the concepts of visibility-aware gating and a streaming cosine-

*Corresponding Author.

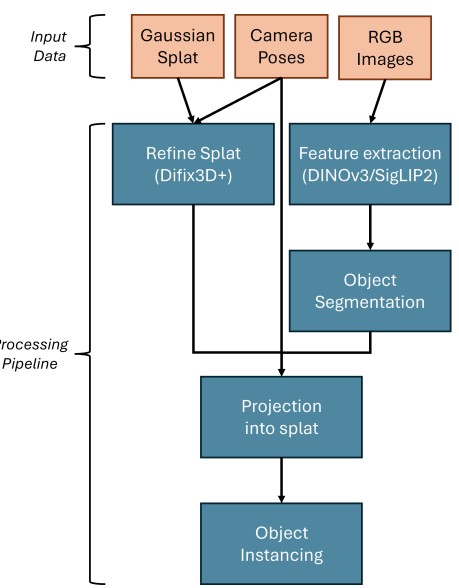

**Figure 2.** Input data and processing pipeline overview.

median feature aggregation to ensure that only truly visible Gaussians receive semantic updates. Complementary to these works, GaussianVLM [7] further shift the focus towards scene-centric reasoning, integrating 3DGS with large VLMs for holistic text-based scene interaction.

Together, these advances provide the methodological foundation for our proposed pipeline, which extends visibility-aware per-Gaussian semantics with explicit object level instancing and inventory generation within complex 3D scenes.

## 3    Methodology

We introduce a five-step pipeline that is able to process objects that are arbitrary both in size and class. The input data are images and their resulting Gaussian Splat with the camera poses. Fig. 2 shows an overview of inputs and the processing pipeline.

1. **Feature extraction.** First, DINOv3 [8] is used to extract a latent space representation of all the input images' features. For performance tests, a comparison set of features will be generated via SigLIP2 [9]. These feature representations are fused and compressed into one single embedding for the whole scene. The advantage of this latent space representation is that it does not rely on distinct classes.

2. **Refine splat.** In the second step, we refine the input Gaussian splat through Difix3D+ [10] to improve 3D reconstruction results and minimize floaters. A sharper splat results in a more precise segmentation in the downstream pipeline.

3. **Object segmentation.** For the third step, the latent space representation is queried with a

search term for an object. If the object is found, all individual input images are searched for the object. From the correlation heatmap we can estimate an individual mask size for segmenting the images utilizing SAMv2.1 [11]. This is important to prevent too large or too small masks that would result in incomplete or too fine-grained segmentations. Especially in crime scene photography one finds overview imagery of the whole scene where for example a blood splatter might be only a small detail in one image but it fills the whole image in an up-close detail shot.

4. **Projection into splat.** The Gaussian splat is then supplemented with the semantics of the object class. By projecting from the input images and their respective camera poses, the individual Gaussians of the object are assigned its class attribute in 3D space. The diverse poses ensure a low number of false negative detections as occlusions are reduced if the scene is captured properly.

5. **Object instancing.** Lastly, we instantiate all found objects and add them to an inventory of the room.

## 4    Conclusion

Pipeline part 1 is set up for the comparison of feature extractors, parts 2, 3 and 4 are fully implemented. The full implementation including part 5 will be presented at Northern Lights Deep Learning 2026. Further, the segmentation step will be updated when v3 of SAM is available.

## Acknowledgments

The authors want to thank the Bavarian State Police (Landeskriminalamt Bayern) for providing datasets of challenging crime scenes for test purposes.

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
