# OpenReview forum: "InV3st: Open-Vocabulary Object Detection and Instance Segmentation in 3D Crime Scenes"
_NLDL.org/2026/Abstracts_Track — NLDL 2026 Abstracts_

### Official Review · Reviewer_MEvE · 2025-10-28

**Soundness:** 3
**Correctness:** 3
**Rating:** 5
**Confidence:** 3

**Summary:**

The paper presents a methodological pipeline designed for detecting and segmenting objects in 3D reconstructions of crime scenes, intended for use in law enforcement. It leverages recently published state-of-the-art techniques to enable open-vocabulary object recognition, allowing the system to identify diverse object classes without predefined constraints. The pipeline is currently under development and not yet fully completed.

**Strengths:**

- Sound motivation
- Methodology is clearly explained and therefore easy to follow.
- Incorporates the latest developments in the field of research.

**Weaknesses:**

- Figure 1 lacks sufficient description and requires revision to better convey its content.
- The authors reference the Bavarian Police’s use of 3D twins since 2012; this claim would be strengthened by citing a published source.
- No (preliminary) results are presented in the paper.
- Part 5 of the Methodology (Object Instancing) is explained too briefly.

---

### Official Review · Reviewer_t2VH · 2025-10-31

**Soundness:** 4
**Correctness:** 3
**Rating:** 5
**Confidence:** 3

**Summary:**

The authors propose a VLM pipeline for open-vocabulary object detection and segmentation within 3D Gaussian Splat reconstructions of crime scenes. The proposed pipeline uses DINOv3 and SigLIP2 for feature extraction to create a unified latent-space representation of the scene that can be queried using natural language. Unlike prior discrete-class detection approaches, this framework enables flexible identification of arbitrary objects in complex and cluttered environments. This is especially useful for analysis of virtual crime scene reconstructions, which often contain large amounts of arbitrary objects.

**Strengths:**

- The authors present a well-described detailed analysis pipeline for their approach.
- The approach of using DINOv3 and SigLIP2 to encode a single latent representation for the scene is a technically sound approach for enabling class agnostic object detection of the whole scene.
- The proposed method of performing 2D segmentation followed by 3D projection is a clever strategy to alleviate computational complexity.
- A clear strength of the paper is enabling open-vocabulary query of scene objects, which is a major limitation of current detection methods.
- The method is general enough to have significance outside the field of crime scene investigation for scene understanding and analysis.

**Weaknesses:**

- The use of several large SOTA methods (DINO, SigLIP2, SAM v2.1, Difix3D+) is likely to make inference of the whole analysis pipeline very computationally costly.
- The number of computational steps in the pipeline may make ablation and hyperparameter tuning difficult.
- The use of 2D segmetation followed by 3D projection may inherit the same limitations of occlusion handling of 2D methods if the scene is not captured from enough angles.
- Inclusion of a few preliminary results of the proposed analysis pipeline would greatly strengthen the correctness of the approach, as the method has many individual parts.

---

### Official Review · Reviewer_un4C · 2025-11-02

**Soundness:** 2
**Correctness:** 2
**Rating:** 2
**Confidence:** 3

**Summary:**

The authors propose InV3st, an open-vocabulary object detection and instance segmentation pipeline for 3D crime scenes from Gaussian splats. The method integrates DINOv3 and SigLIP2 for feature extraction, combines them with SAM-based segmentation. To reduce computational cost, the system operates in 2D image space and segments objects within the 3D scene. The approach aims to assist forensic investigation by automating object identification in complex, cluttered 3D environments. However, the abstract states that results and evaluation will be presented at the conference, meaning no quantitative or qualitative outcomes are included at this stage.

**Strengths:**

The topic is timely and relevant, combining 3D scene understanding with vision-language models.
The potential application to forensic investigation is original and socially significant. The abstract clearly describes the methodological pipeline.

**Weaknesses:**

While the topic is interesting and the proposed pipeline is clearly described, the submission would benefit from greater completeness and contextual grounding. At this stage, no qualitative or quantitative results are reported, which makes it difficult to engage in an informed discussion during the poster session. The introduction could be strengthened by including references, and a brief reflection on the ethical and societal implications of applying AI methods in forensic contexts would add important depth. Additionally, Figure 1 would be more informative with a descriptive caption. Overall, the work appears to be at a conceptual or proposal stage.

---

### Decision · Program_Chairs · 2025-11-05

**Decision:**

Accept

**Comment:**

The abstract is of interest to the community and should be presented at the conference.